# A New Agricultural Drought Disaster Risk Assessment Framework: Coupled a Copula Function to Select Return Periods and the Jensen Model to Calculate Yield Loss

**Hongjun Lei** [1], **Jie Yu** [1], **Hongwei Pan** [1,*], **Jie Li** [2], **Shah Jahan Leghari** [3], **Chongju Shang** [4], **Zheyuan Xiao** [1], **Cuicui Jin** [1] and **Lili Shi** [1]

1  School of Water Conservancy, North China University of Water Resources and Electric Power, Zhengzhou 450046, China
2  China Machinery Industry International Cooperation Co., Ltd., Zhengzhou 450018, China
3  College of Land Science and Technology, China Agricultural University, Beijing 100193, China
4  Guizhou Water Conservancy Research Institute, Guiyang 550002, China
*  Correspondence: panhongwei@ncwu.edu.cn; Tel.: +86-155-1615-1051

**Abstract:** China is one of the regions with the most frequent drought disasters and serious social and economic losses. Agricultural drought is the most serious natural disaster. Due to climate change, the regional agricultural drought risk assessment has always been the focus of the academic circle. This study takes Zunyi City as an example, which is the most typical city of karst landform development. The monthly precipitation data set of ground meteorological observation stations in Zunyi City from 1956 to 2020 was selected, and the drought characteristic variables were extracted by the coupled use of the precipitation anomaly percentage (*Pa*) index and the theory of runs. A copula function was applied to establish the joint distribution model of characteristic variables, obtaining the drought frequency and drought return periods. Combined with the Jensen model, the agricultural drought loss rate under different drought return periods in the target year (2020) was calculated and evaluated. The results showed that the Gumbel-Hougaard copula function was suitable for the joint distribution of drought joint variables in Zunyi City. From 1956 to 2020, fewer droughts occurred in Zhengan and Wuchuan, and the most droughts took place in Fenggang, Meitan, and Yuqing. The average drought duration in each county was about 1.5 months, and the average drought severity was about 0.35 in spatial distribution. Crop loss rate caused by drought increased and the affected area expanded with the increase of drought return periods (5, 10, 20, 50, and 100 years) in temporal distribution. Meanwhile, the drought disaster was most drastic in the eastern region, followed by the south, north, west, and central area. The results were highly consistent with the historical drought in Zunyi City, which verified the validity of the model. This study could provide scientific knowledge for drought resistance and reasonable mitigation programing for the security of the regional agricultural production and the sustainability of social and economic development.

**Keywords:** theory of runs; copula function; Jensen model; drought risk; Zunyi City

## 1. Introduction

Drought disaster is one of the most frequent, severe natural disasters widespread on a year-to-year basis [1]. In order to grasp the overall situation of drought disaster risk and enhance the comprehensive ability of the whole society to prevent and resist drought disasters, a national drought disaster risk assessment was carried out according to the requirements of the Implementation plan of the First National Survey of Comprehensive Natural Disaster Risks in China ((2020) No. 2). According to the classification standard of the World Meteorological Organization, definitions of drought are clustered to four types: meteorological, agricultural, hydrological, and socio-economic [2,3]. Among them, meteorological drought is the main cause of agricultural drought. Agriculture was a direct

and sensitive sector of meteorological drought [4,5]. Drought has reduced China's grain production by 16,159 million ha in the last 60 years [6]. Investigating agricultural drought risk has become the focus of scientific research. Many scholars used meteorological drought indicators to characterize agricultural drought [7,8]. The precipitation anomaly percentage (*Pa*) index could easily directly reflect the drought caused by abnormally reduced precipitation in Guizhou [9]. It is widely used in drought assessment because of its convenience in data acquisition, simple calculation, and clear physical significance.

Currently, most drought disaster risk assessment methods evaluate drought from four aspects: hazard, exposure, vulnerability, and drought resistance [10–12]. These methods contain many evaluation indexes and need a large amount of statistical data, and the reliability of the data greatly constrains the accuracy of the results. For research on crop yield loss, the statistical survey of post-drought disaster is used [13,14]. It is difficult to accurately estimate drought loss due to the delay of disaster information. The drought returns period is an important indicator used to evaluate the severity of drought. Drought is characterized by continuity, and some droughts last for several years or occur several times in one year. The reciprocal of design flood frequency is often used to calculate the returns period, and is not suitable for the returns period analysis of drought events. Thus, the multi-dimensional joint distribution function of drought duration and severity characteristic variables is constructed by using a copula function to calculate the return periods of drought events at present [15–17]. The Jensen model was often used to establish a dynamic assessment on the agricultural drought risk by connecting rainfall, irrigation, and leakage when crop water shortage occurs in various growth periods. It could be used to characterize the effect of drought on crop yield loss and to resolve the relationship between them. By the coupled use of a copula function and the Jensen model, as well as logistic equation fitting with meteorological droughts [18], the relationship between meteorological droughts and agricultural droughts could be resolved and could thus be proactively used to prevent and cope with risks, ensuring regional agricultural production and regional sustainability.

Southwest China is dominated by karst landscapes and is affected by the irregular monsoon climate and large differences in terrain height, resulting in the most drought-prone region in China, which poses a serious threat to agriculture. Since the beginning of the 21st century, the frequency of drought in Southwest China has increased significantly. In 2006, the drought frequency in Chongqing, Sichuan and other places in Southwest China occurred at 100 years of return period. From 2009 to 2010, the most severe autumn–winter–spring drought since the beginning of meteorological records occurred in Southwest China. The drought area reached $8 \times 10^6$ hm$^2$, resulting in a yield loss of different crops in Yunnan, Guizhou, and northwestern Guangxi [19]. The frequency of drought in southern Sichuan, Yunnan, and western Guizhou reached 50% in 2011–2014 [20]. In the future, the risk of agricultural drought might increase due to the influence and restriction of various environmental factors, food demand, disaster resistance, and other factors. The purpose of this study is to use a copula function to establish a joint distribution model of drought characteristic variables, obtain drought risk frequency and returns period, and calculate agricultural drought losses based on the Jensen model. By the coupled use of a copula function and the Jensen model, a regional agricultural drought risk assessment model with multiple returns periods is constructed, which provides a new idea for regional agricultural drought risk assessment.

## 2. Materials and Methods

### 2.1. Description of the Study Area

Zunyi City is located in the north of Guizhou Province (105°36′ E–108°13′ E, 27°08′ N–29°12′ N), has a subtropical monsoon climate, a complex topography, obvious vertical differences, and average annual rainfall between 800 and 1300 mm [21]. Zunyi was known as the "granary of northern Guizhou", with grain production accounting for about one four of the total yield production in Guizhou Province, and it is an important grain production

base. Drought is the main meteorological factor influencing the local crop yield. The geographical location of the study area is shown in Figure 1.

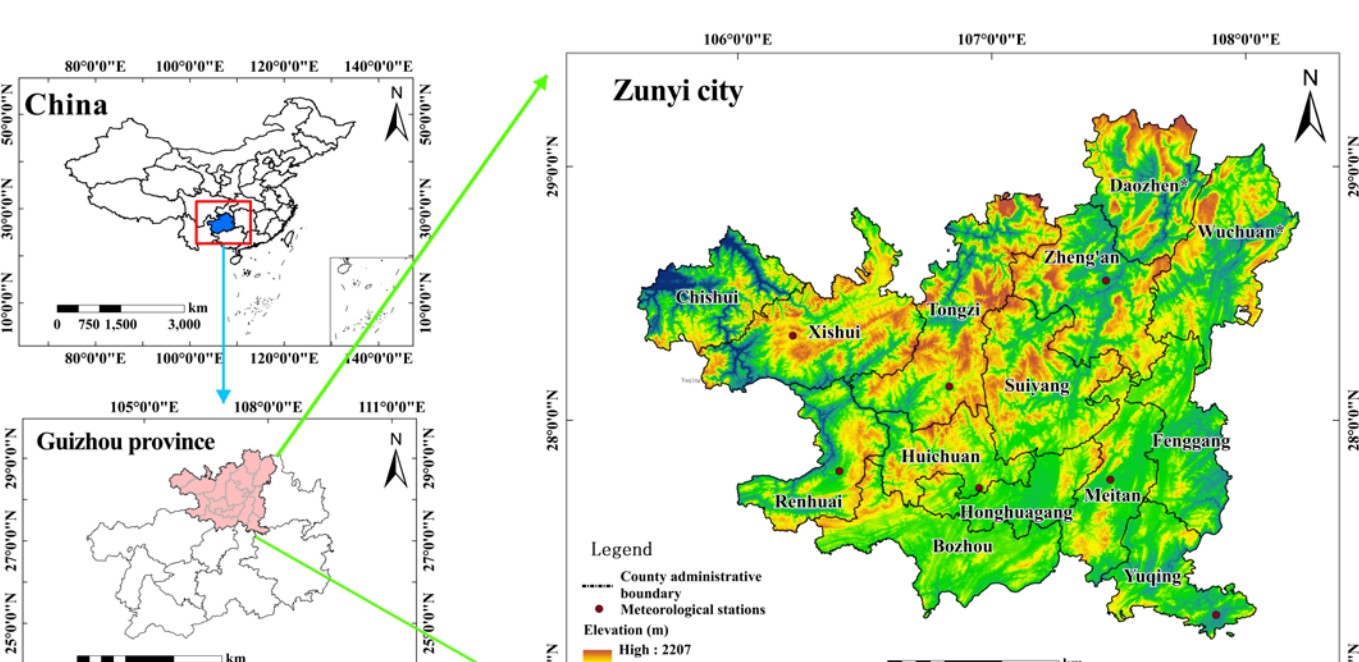

**Figure 1.** Geographical location of the study area. Note: * indicates the abbreviation of an autonomous county.

### 2.2. Data Sources

The daily precipitation data and meteorological data of seven surface meteorological observation stations in Zunyi from 1956 to 2020 were acquired from the China Meteorological Data Service Center (http://data.cma.cn (accessed on 15 April 2022)). DEM data were obtained from the Geospatial Data Cloud (https://www.gscloud.cn (accessed on 19 April 2022)) with a resolution of 30 m. The irrigation area in 2020 was extracted from the Zunyi Statistical Yearbook (2021). The resolution of drought characteristic variables and the joint analysis of univariate and bivariate were calculated in MatlabR2018b (MathWorks, Natick, MA, USA). Histograms and curve charts are plotted using the Origin 2021 program (OriginLab Corporation, Northampton, MA, USA). The spatial distribution map of drought disaster loss was illustrated using ArcGIS10.1 (GeoScene, Beijing, China).

### 2.3. Evaluation Object Selection

Rice is the main food crop in Guizhou Province and one of the high-yield crops. The annual planting area is about 670,000 hm$^2$, accounting for about 25% of the cultivated area of food crops, and the yield accounts for 30% of the total grain [22]. According to the analysis of historical drought events, rice yield is the most affected by drought disasters. In the local area, rice is the representative crop for the assessment of agricultural drought losses. Therefore, this paper takes rice as the reference to analyze the spatial distribution of agricultural drought risk.

*2.4. Methodology*

2.4.1. Drought Classification

Based on the meteorological data between 1956 and 2020 in Guizhou Province [23,24], *Pa* was selected for analysis as the drought indicator in this study. The criteria for drought classification are presented in Table 1. The calculation formulas are given below:

$$Pa = \frac{P - \overline{P}}{\overline{P}} \times 100\%$$ (1)

and

$$\overline{P} = \frac{1}{n} \sum_{i=1}^{n} P_i$$ (2)

where $P$ is the precipitation amount for a certain time period; $\overline{P}$ is the precipitation climate average; and $n$ is the number of samples.

**Table 1.** Drought classification based on *Pa*.

| Drought Grade | Monthly Scale *Pa* Value (%) |
|---|---|
| Normal | $-40 < Pa$ |
| Light drought | $-60 < Pa \leq -40$ |
| Moderate drought | $-80 < Pa \leq -60$ |
| Severe drought | $-95 < Pa \leq -80$ |
| Extreme drought | $Pa \leq -95$ |

Note: derived from *Guizhou Drought Standard* DB 52/T 1030, 2015 [25].

2.4.2. Analysis of Drought Feature Variables

The drought stress during 1956–2020 was identified according to the theory of runs and the characteristic variables of drought duration, and drought severity was calculated. The specific steps of drought event identification are shown in Figure 2 [26]. The threshold value of the drought index was selected referring to the specification document of the *Guizhou Drought Standard* (DB52T 1030, 2015 (accessed on 2 May 2022)), where drought was likely to occur when there was a gap in the precipitation series, so the first truncation level was set as $R_0 = 0$; the recommended values of light drought level were taken as drought event identification thresholds: $R_1 = -0.4$ and $R_2 = -0.6$, respectively.

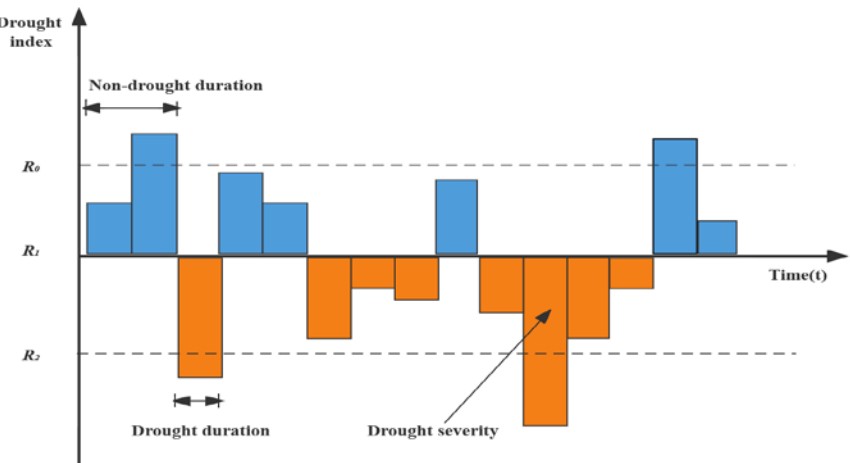

**Figure 2.** Drought process identification and drought characteristic variable extraction.

2.4.3. Copula Function for Drought Frequency and Return Period

It is difficult to reflect drought impact comprehensively by frequency analysis from only one characteristic variable of drought, so it is necessary to establish a joint distribu-

tion function for drought frequency. The marginal distribution function of the drought characteristic variable was determined first as:

$$F(x) = \frac{\beta^{\alpha}}{\Gamma(\alpha)} (x - a_0)^{\alpha - 1} e^{-\beta(x - a_0)} \tag{3}$$

where $x$ is the statistical parameters, such as drought duration, drought severity; $\alpha$, $\beta$, and $a_0$ are the distribution parameters by using the maximum-likelihood method [27].

The symmetric Archimedean copula function is currently the key method for resolving bivariate joint probability distributions. There are three types of Archimedean copula functions that are most commonly used (Table 2). The root means square error (RMSE) was used to test the fit degree [28].

**Table 2.** Archimedean copula function type.

| Archimedean Copula | Function Expressions | Parameter Value |
|---|---|---|
| Gumbel-Hougaard (G-H) | $F_{D,S}(d,s) = \exp\left\{ -[(-lnF_D(d))^{\theta} + (-lnF_S(s))^{\theta}]^{1/\theta} \right\}$ | $\theta > 1$ |
| Clayton copula (C-C) | $F_{D,S}(d,s) = (F_D(d)^{-\theta} + F_S(s)^{-\theta} - 1)^{-1/\theta}$ | $\theta \geq 0$ |
| Frank copula (F-C) | $F_{D,S}(d,s) = -\frac{1}{\theta} ln[1 + \frac{(e^{-\theta F_D(d)} - 1)(e^{-\theta F_S(s)} - 1)}{e^{-\theta} - 1}]$ | $\theta \in R$ |

Note: $F_D(d)$ is the drought duration marginal distribution function; $F_S(s)$ is the drought severity marginal distribution function; $\theta$ is a parameter used in the Kendall correlation coefficient ($\tau$), $\theta = 1/(1 - \tau)$ [29].

Drought frequency was derived from the joint probability distribution between drought duration and drought severity [30]:

$$P = 1 - F_D(d) - F_S(s) + F_{D,S}(d,s) \tag{4}$$

Compared with the frequency of drought, drought return periods was used to reflect the uncertainty of drought. It was characterized by image and easy to understand, and it was conducive to the release of the drought situation in the actual drought mitigation work [30]:

$$T_{DS} = \frac{E(L)}{P(D \geq d \, and \, S \geq s)} = \frac{E(L)}{1 - F_D(d) - F_S(s) + F_{D,S}(d,s)} \tag{5}$$

$$T_{DS} = \frac{E(L)}{P(D \geq d \, or \, S \geq s)} = \frac{E(L)}{1 - F_{D,S}(d,s)} \tag{6}$$

where $E(L)$ is the expected value of drought interval.

### 2.4.4. Jensen Model for Yield Loss Calculation

According to the identification of drought events, the daily precipitation and evaporation during the drought events were selected, and the meteorological data in 2020 were chosen for the other non-drought periods. The Penman–Monteith equation was used to calculate the daily $ET_0$ [29], and the division of rice growth stages was coupled with the daily precipitation in each county-level administrative region of Zunyi City. The crop water demand coefficients used by local water conservancy departments are presented in Table 3. According to the effective irrigated area, the site-specific crop irrigation quota was derived from the local standard of water quota in Guizhou Province (DB52/T 725, 2019) [31]. The irrigation water allocation was calculated based on the irrigation water utilization coefficient (Table 4). The crop irrigation water was allocated to each rice growth period and the improved Jensen model was used for drought loss calculation.

**Table 3.** Water requirement coefficient during the rice growth period.

| Growth Stage | Rejuvenation Stage | Tillering Stage | Jointing and Booting Stage | Heading and Flowering Stage | Grain Filling Stage | Mature Stage |
|---|---|---|---|---|---|---|
| Growth period | 5.20–6.01 | 6.02–7.06 | 7.07–8.04 | 8.05–8.19 | 8.20–9.03 | 9.04–9.26 |
| $\alpha$−value | 1 | 1 | 1 | 1.57 | 0.9 | 0.9 |

**Table 4.** Irrigation water utilization coefficient in 2020.

| County Name | Utilization Coefficient of Irrigation Water | County Name | Utilization Coefficient of Irrigation Water | County Name | Utilization Coefficient of Irrigation Water |
|---|---|---|---|---|---|
| Honghuagang | 0.492 | Zheng'an | 0.485 | Yuqing | 0.487 |
| Huichuan | 0.489 | Daozhen * | 0.484 | Xishui | 0.483 |
| Bozhou | 0.492 | Wuchuan * | 0.481 | Chishui | 0.487 |
| Tongzi | 0.486 | Fenggang | 0.491 | Renhuai | 0.485 |
| Suiyang | 0.480 | Meitan | 0.491 | | |

Note: * indicates the abbreviation of an autonomous county.

The simplified Jensen model was adopted as described by Liu et al. [32] and Li et al. [33] to calculate the yield drought loss. Because of the ignorance of the drainage and leakage, the calculation was not quite in line with the actual situation. Thus, the model was improved by considering the paddy field leakage and the thickness difference in the water layer between the beginning and the end of the growth stage:

$$\frac{Y_a}{Y_m} = \prod_{i=1}^{n}\left(\frac{ET_{ai}}{ET_{mi}}\right)^{\lambda_i} = \prod_{i=1}^{n}\left(\frac{P_i + W_{gi} - L_i - n}{ET_i}\right)^{\lambda_i} \tag{7}$$

where $P_i$ is the rainfall depth during the rice growth period (mm); $W_{gi}$ is the irrigation water during the time interval (mm); $L_i$ is the seepage during the time period, 2 mm/d [22]; $ET_i$ is the field water requirement during each growth stage by using the $\alpha$-value method (mm) [34]; and $\lambda_i$ are the sensitivity coefficients in various growth stages [35].

Crop yield loss rate caused by drought was calculated by:

$$Id_i = \left[1 - \prod_{i=1}^{n}\left(\frac{P_i + W_{gi} - L - n}{ET_i}\right)^{\lambda_i}\right] \times 100\% \tag{8}$$

Allocation of the irrigation amount during the rice growth period was calculated by [36]:

$$W_{gi} = \frac{m_i \cdot A_i}{\sum_{i=1}^{n}(m_i \cdot A_i)} \times W_n \times \eta \tag{9}$$

where $m_i$ is the crop irrigation quota (m³/hm²); $A_i$ is the effective crop irrigation area, (hm²); $W_n$ is the irrigation water consumption of farmland (million m³); $n$ is the number of crop species (five major crops were included for irrigation water allocation, i.e., wheat, corn, rape, and roasted tobacco in Zunyi City); and $\eta$ is the irrigation water utilization coefficient.

## 3. Results

### 3.1. Agricultural Drought Disaster Characteristics

3.1.1. Analysis of Drought Characteristics

The drought events during 1956–2020 in each county-level administrative region of Zunyi City were identified based on the monthly Pa and the characteristic variables of drought events were extracted using theory of runs. The statistical results are presented in Table 5. From 1956 to 2020, fewer droughts occurred in Zhengan and Wuchuan, and most droughts took place in Fenggang, Meitan, and Yuqing. The average drought duration in each county was about 1.5 months, and the average drought severity was about 0.35.

**Table 5.** Statistical results of drought characteristic variables.

| County Name | Number of Droughts (Times) | Average Drought Duration (Months) | Average Drought Severity | Maximum Drought Duration (Months) | Maximum Drought Severity |
|---|---|---|---|---|---|
| Honghuagang | 55 | 1.78 | 0.37 | 6 | 1.02 |
| Huichuan | 54 | 1.61 | 0.33 | 5 | 0.74 |
| Bozhou | 58 | 1.66 | 0.39 | 5 | 1.18 |
| Tongzi | 26 | 1.54 | 0.30 | 3 | 0.66 |
| Suiyang | 48 | 1.58 | 0.33 | 3 | 0.66 |
| Zheng'an | 21 | 1.57 | 0.30 | 3 | 0.47 |
| Daozhen * | 26 | 1.58 | 0.43 | 3 | 0.66 |
| Wuchuan * | 21 | 1.48 | 0.30 | 3 | 0.47 |
| Fenggang | 71 | 1.82 | 0.39 | 5 | 0.87 |
| Meitan | 71 | 1.80 | 0.39 | 5 | 0.87 |
| Yuqing | 71 | 1.80 | 0.39 | 5 | 0.87 |
| Xishui | 33 | 1.45 | 0.34 | 6 | 1.02 |
| Chishui | 43 | 1.58 | 0.33 | 6 | 1.24 |
| Renhuai | 27 | 1.41 | 0.30 | 3 | 0.69 |

Note: * indicates the abbreviation of an autonomous county.

### 3.1.2. Establishment of Two-Dimensional Joint Distribution

Table 6 indicated that in the 14 county-level administrative regions of Zunyi City, the best fitting formula was the Gumbel-Hougaard copula (G-H), with 93% accuracy, and the best fitting effect was that of the Frank copula (F-C) with, 7% accuracy. The Clayton copula (C-C) is not applicable to Zunyi City. Therefore, the G-H function was selected for the joint distribution of drought joint variables in Zunyi City.

**Table 6.** Copula function goodness of fit evaluation and optimal function.

| County Name | G-H | C-C | F-C | Optimal Copula Function | Parameters θ |
|---|---|---|---|---|---|
| Honghuagang | 0.2475 | 0.4280 | 0.3015 | G-H | 1.6618 |
| Huichuan | 0.3028 | 0.4144 | 0.3326 | G-H | 1.3159 |
| Bozhou | 0.2815 | 0.5036 | 0.3486 | G-H | 1.6814 |
| Tongzi | 0.1729 | 0.2156 | 0.1989 | G-H | 1.0844 |
| Suiyang | 0.1566 | 0.1816 | 0.1511 | F-C | 0.8570 |
| Zheng'an | 0.1380 | 0.2191 | 0.1697 | G-H | 1.2887 |
| Daozhen * | 0.1601 | 0.3152 | 0.2009 | G-H | 1.4999 |
| Wuchuan * | 0.1675 | 0.2473 | 0.2081 | G-H | 1.2723 |
| Fenggang | 0.2673 | 0.4379 | 0.3187 | G-H | 1.4925 |
| Meitan | 0.2735 | 0.4460 | 0.3238 | G-H | 1.5044 |
| Yuqing | 0.2895 | 0.4547 | 0.3386 | G-H | 1.4950 |
| Xishui | 0.2149 | 0.3170 | 0.2773 | G-H | 1.2532 |
| Chishui | 0.1909 | 0.3708 | 0.2895 | G-H | 1.3769 |
| Renhuai | 0.1453 | 0.2049 | 0.1759 | G-H | 1.0851 |

Note: G-H indicates Gumbel-Hougaard copula; C-C indicates Clayton copula; F-C indicates Frank copula. * indicates the abbreviation of an autonomous county.

The joint probability fit of drought is shown in Figure 3. The joint probability distribution was consistent with the empirical frequency distribution, indicating that the bivariate joint probability distribution constructed with the G-H function was reasonable. With the increase in drought duration or drought severity, the cumulative value of the joint probability showed a rapid rise. When drought severity was greater than 0.5 and drought duration was greater than 2 months, the cumulative value of the joint probability showed a slowly increasing trend. In general, drought events in Zunyi City were characterized by high-severity in short-duration cases and long-duration, low-severity cases. The change rate of the joint cumulative probability gradually decreases with the increase in drought duration and drought severity.

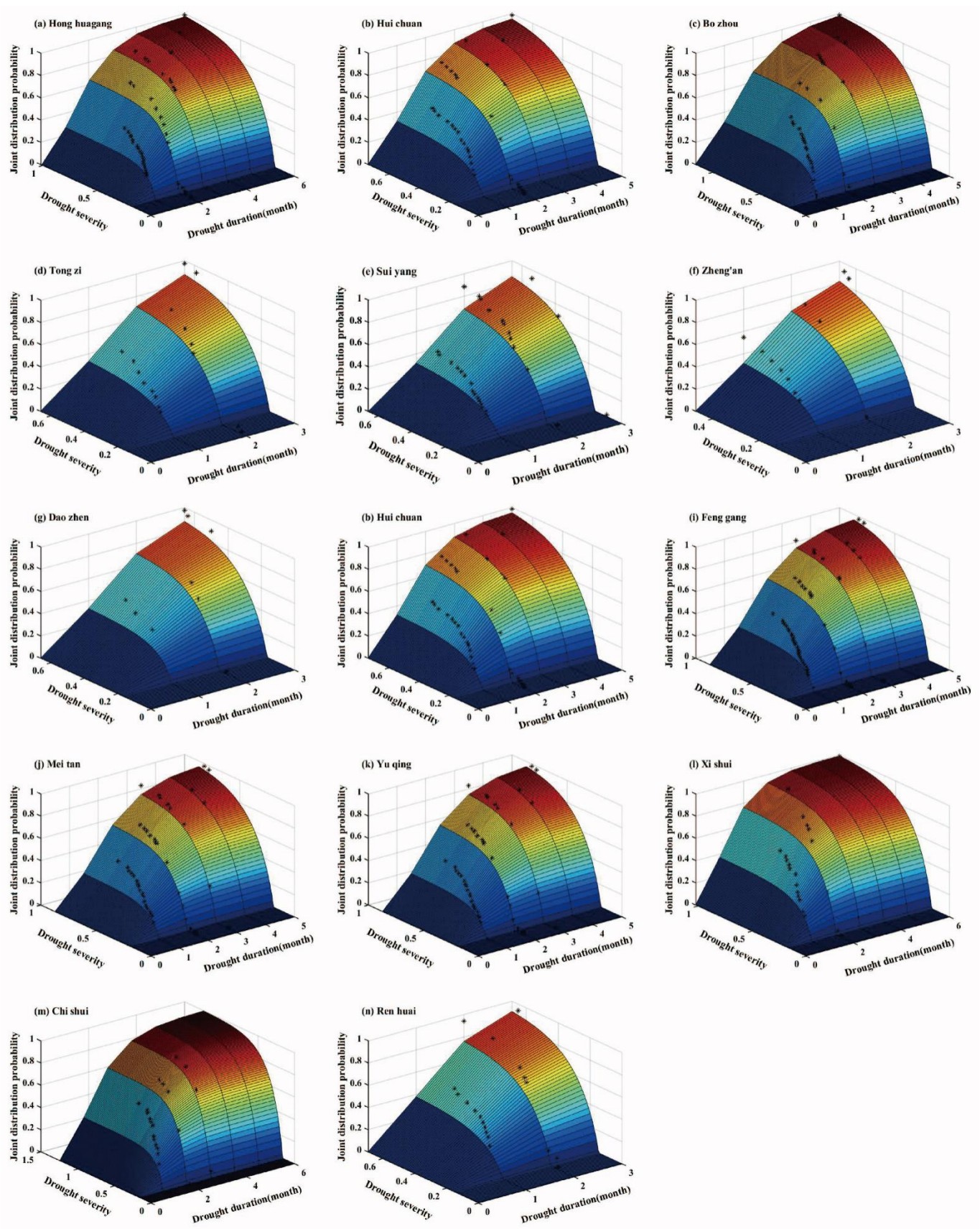

**Figure 3.** Joint probability distribution of drought duration and drought severity. Note: * empirical joint probability distribution points.

### 3.1.3. Rational Analysis of Drought Characteristics Results

According to the statistical data of the *China Meteorological Disasters Dictionary-Guizhou Volume* [37], the main drought events in Guizhou Province occurred in 1963, 1966, 1978, 1988, 1990, and 1992. Among them, severe drought occurred from March to mid-July in 1963, with a total area of 453 khm$^2$ in spring and summer, and 299 khm$^2$ in Guizhou Province. According to the records of the Guizhou Flood and Drought Disaster Monograph Continuation [38] and the Guizhou Drought Disaster Risk Assessment and Climate Prediction [23], there were three years of extreme drought between 2000 and 2013, with 2010, 2011, and 2013 being the most severe. Drought hit the hardest in 2011, affecting 659 ha, followed by 493 ha in 2013 and 447 ha in 2010.

Based on the yearly drought event identification, the cumulative values of drought severity in the district of Zunyi City from 1956 to 2020 are plotted in Figure 4. Before 2000, the cumulative severity of drought in 1963, 1978, 1988, and 1992 in Zunyi City was higher than that in other years. Thereafter, the cumulative drought severity of Zunyi in 2010, 2011, and 2013 was different. These drought years coincided with the historical data, indicating that the selection of drought indicators and the division of thresholds are suitable for the research area.

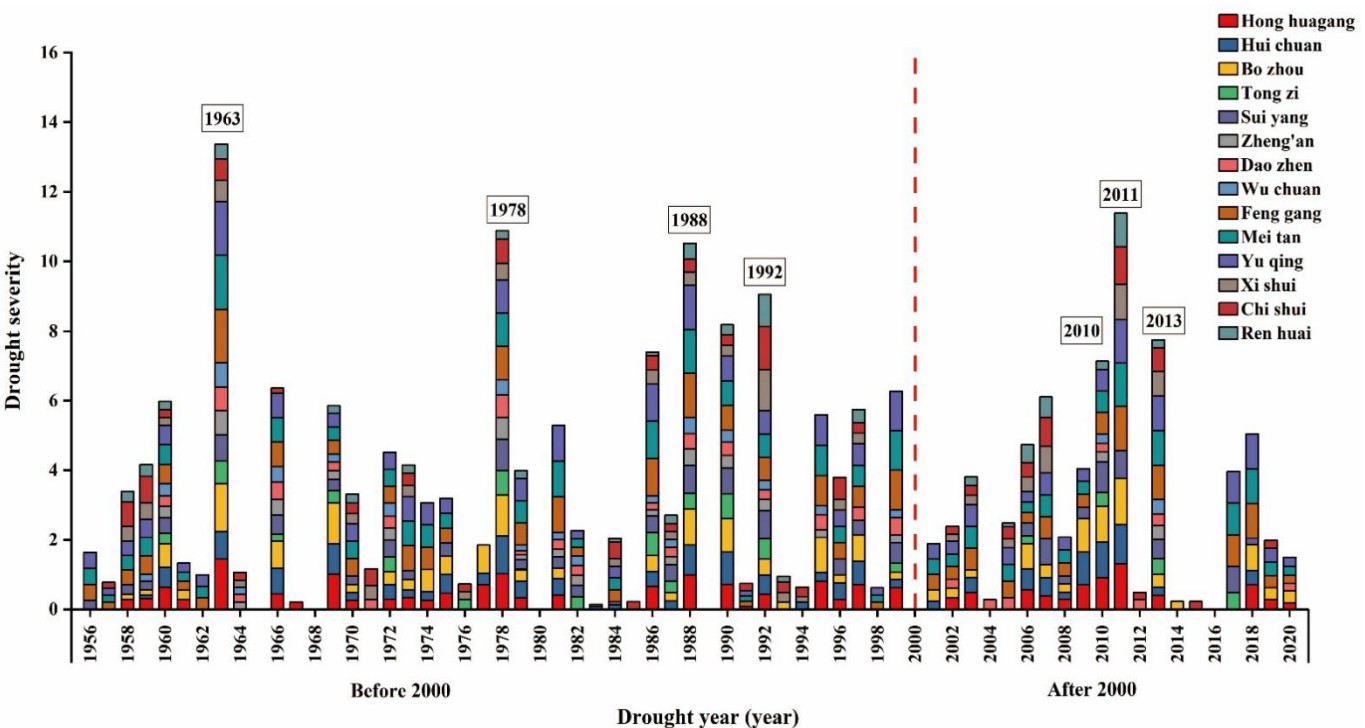

**Figure 4.** Cumulative severity of drought in Zunyi City from 1956 to 2020.

### 3.2. Agricultural Drought Risk Assessment

#### 3.2.1. Distribution Characteristics of Irrigation Water

According to the Guizhou Provincial Water Resources Bulletin, the total amount of water resources in 2018 (a normal year) was 7.8% lower than the multi-year average. In 2020, the total amount of water resources was abundant, so it was a wet year. Therefore, the amount of rice irrigation water in Zunyi City was calculated with 2018 as the base year, and the distribution characteristics are shown in Figure 5. The amounts of rice irrigation in Honghuagang County, Huichuan County and Renhuai County were the largest, while the irrigation amounts in Wuchuan County, Daozhen County, Fenggang County, Meitan County, and Suiyang County were insufficient. The overall distribution of irrigation water was more in the middle and less in the east, mainly affected by topography and land types. Honghuagang, Huichuan, and other counties are located in the low hilly and wide valley

basin area, with relatively small surface fluctuation (about 100−200 m). The mountain is not very high, the terrain is open and flat, the cultivated land is concentrated and contiguous with more fields and less soil, it has good irrigation and farming conditions, and a high degree of land use. Fuchuan County, Daozhen County, and other counties are located in mountainous and canyon areas, with high terrain in the south and low terrain in the north, and high mountains and deep valleys. The altitude is about 900−1500 m [39], the relative height difference is 600 m, the cultivated land is scattered, and there are few dams, resulting in less irrigation water.

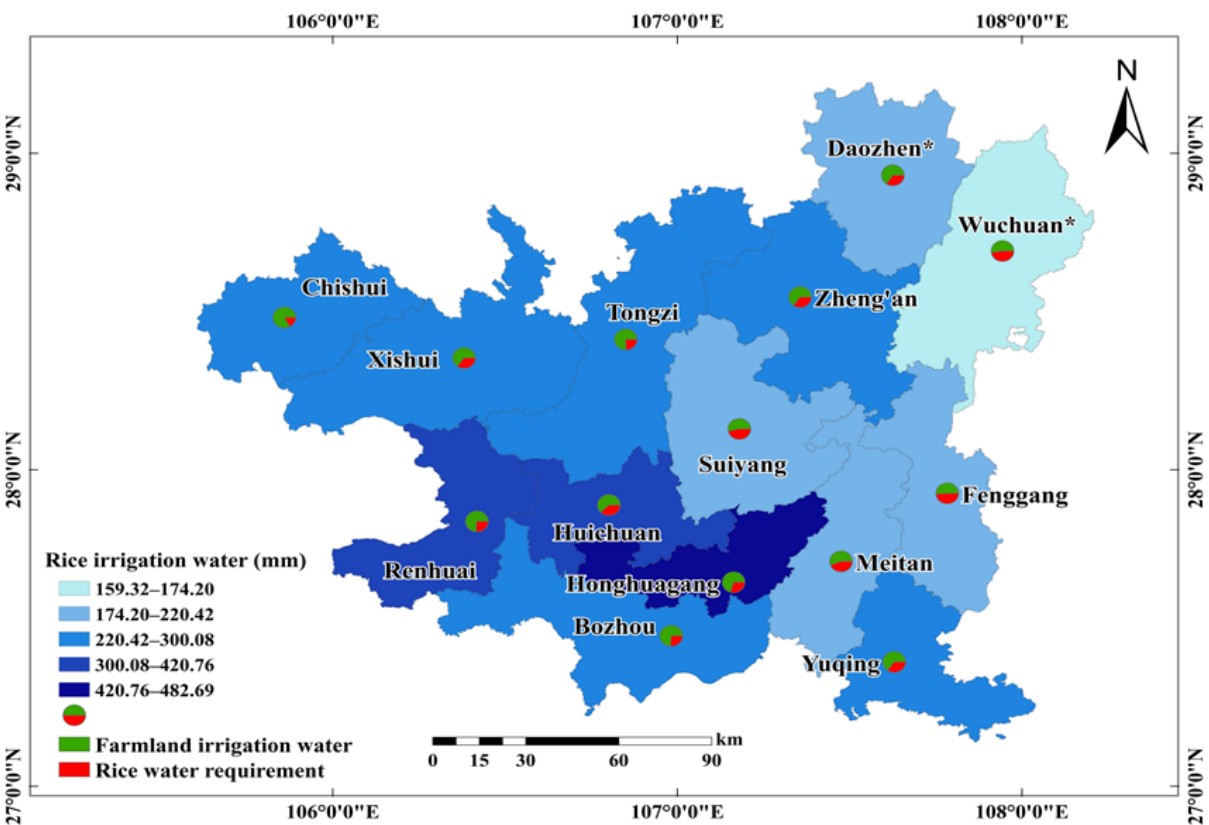

**Figure 5.** Distribution characteristics of irrigation water amounts for rice. Note: * indicates the abbreviation of an autonomous county.

### 3.2.2. Drought Frequency and the Agricultural Drought Loss Rate

The growth period of rice in Zunyi City was from May 20 to September 26. The frequency of all drought events occurring in this period was selected for each county and district, the agricultural drought loss rate in the corresponding period was calculated, and the risk curve of drought frequency–agricultural drought loss was established by logarithmic equation fitting (Figure 6). The drought frequency and drought loss rate have a good correlation, the scatter points are evenly distributed on both sides of the fitted curve, and the correlation coefficients of the fitted equations are higher than 0.5. The drought loss rate shows an increasing trend as the drought frequency decreases, which is in line with the objective reality.

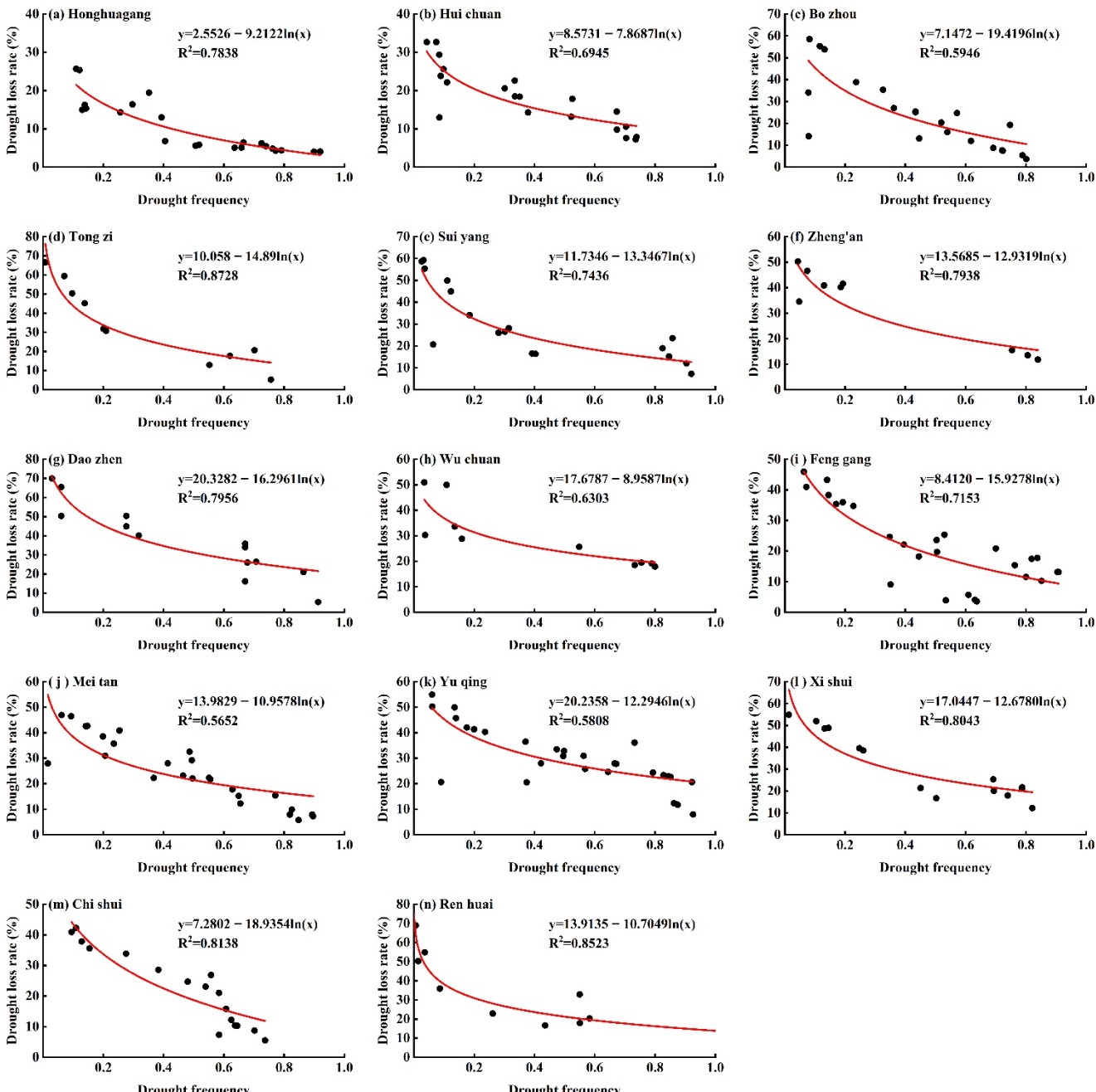

**Figure 6.** Risk curve of agricultural drought disaster loss in each county.

### 3.2.3. Spatial Distribution of Agricultural Drought Loss

According to the fitting formula, the drought loss rates corresponding to different drought frequencies were calculated and the drought frequency was converted into drought return periods. The spatial distributions of drought losses under different drought return periods were derived (Figure 7).

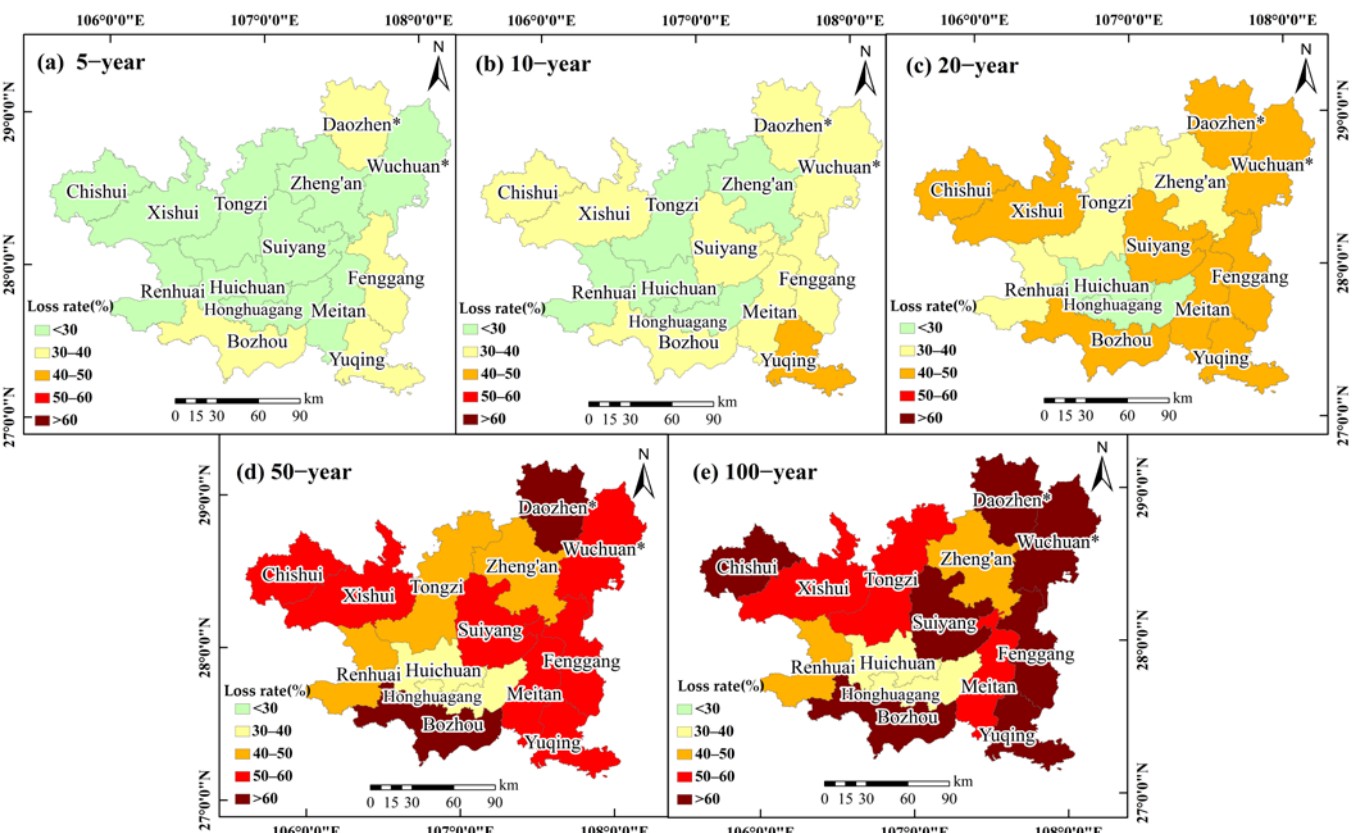

**Figure 7.** Spatial distribution of agricultural drought disaster loss.

The drought loss rate at different return periods of Zunyi City showed a trend of light in the middle and heavy in the east. During the 5-year return period (Figure 7a), the agriculture drought loss rate ranged approximately from 11.88 to 36.64%, with an average of 23.02%; during the 10-year return period (Figure 7b), the agriculture drought loss rate was approximately 17.98–42.34%, with an average of 30.64%; the agriculture drought loss rate was 23.90–49.80% during the 20-year return period, with an average of 39.19% (Figure 7c); 31.84–64.02% during the 50-year return period, with an average of 50.48% (Figure 7d); and 37.40–75.27%, with an average of 59.03%, during the 100-year return period (Figure 7e). Except for the 5-year return period, the agriculture drought loss rates in the remaining recurrence periods (10, 20, 50, and 100 years) were largest in Daozhen, Wuchuan, Fenggang, Meitan, Yuqing, Bozhou, Suiyang, Xishui, and Chishui, and smallest in Honghuagang and Huichuan. The drought disaster is most severe in the eastern region, followed by the south, north, west, and central.

Zunyi City is influenced by atmospheric circulation and in the summer, the western Pacific subtropical high-pressure flow extends westward and northward to control the eastern part of the region, resulting in a heavy drought in the east and a light drought in the west [40,41]. The regional distribution of water resources is very unbalanced, with more water in the west and less in the east, abundant in the north and less in the south, and more water in the flood season and less in the dry season. In addition, because of the karst landform, heavy surface leakage, thin soil, and poor water retention in this area, drought is highly likely to occur [42]. According to the investigation report of drought disaster in Zunyi City, Honghuagang and Huichuan districts are located in the central part of Zunyi City and the drought degree was relatively light because of low latitude, weakened cold air coming from Siberia southward, a developed tributary water system, and abundant water supplied by multiple water sources. Some areas such as Daozhen, Wuchuan, and Fenggang are not equipped with emergency backup water sources, and some areas such as Meitan and Tongzi are supplied with water via a single source, so the drought emergency response

capability is relatively weak. There are a lot of sloping soils in Bozhou and other areas and the land surface has little forest cover, so the soil has poor water retention capacity. In addition, the surface water system is not well developed, with less distribution of water conservancy projects in the area. And the arable land and concentrated living areas are more widespread so there is a great water demand, resulting in frequent yearly drought disaster. The eastern and southern regions are prone to drought and should be prioritized in disaster prevention and mitigation strategies.

## 4. Discussion

### 4.1. Frequent Drought Disaster Areas and Cause Analysis

Drought is the most complex natural phenomenon. Since the beginning of the 20th century, with the intensification of global warming, the scope and frequency of droughts have increased. In this paper, through the identification of drought events in various counties and districts of Zunyi City in the past 65 years, it was found that there were 71 drought events in Fenggang, Meitan and Yuqing, followed by Bozhou, which had 58 drought events. The average drought intensity in Bozhou was 0.39, which was an area prone to drought disaster. There were droughts in Daozhen County, but the average drought intensity was the highest, and it was the county with the most severe drought. This is consistent with the results of drought management in key counties in the study of Joshan [43]. The frequent drought in Guizhou Province is due to its unique geographical location. Guizhou province is located in the hinterland of the karst region in southwest China, and the karst outcropping area accounts for 61.9% of the total land area in Guizhou Province; its unique geographical location and topographic characteristics make it easy for it to suffer from droughts. On the one hand, the special lithology and complex geological structure of the karst canyon area caused strong vertical development of landforms, obvious vertical differentiation of the regional landscape, and continuous undulating topography. Due to the large terrain drop, the thin soil, and poor water retention capacity, the surface water retention capacity and regulation capacity are low, and the rainfall quickly flows into the ground along the fissures and slopes without surface vegetation cover [44]. Furthermore, water vapor mainly comes from the westerly or southwesterly airflow brought by the cyclonic circulation in the Bay of Bengal and the southerly airflow advances along the western side of the subtropical high pressure in the South China Sea region, but the northerly airflow on the eastern side of the Bay of Bengal anticyclone and the northeasterly airflow in the northern part of the South China Sea cyclone block the water vapor channels that are transported to Guizhou all year round [45], resulting in less local water vapor and precipitation. The water vapor flux is dispersed in most areas, resulting in abnormally low precipitation in Guizhou and thus persistent drought events.

### 4.2. Advantages of the Drought Risk Assessment Method Proposed in this Study

Crop yield losses caused by drought depend on both the amount of rainfall and the water consumption of rice during various growth stages. Current agricultural drought research mainly focuses on the effect of rainfall events on drought occurrence. Not enough attention has been paid to the effect of soil and crop on drought during agricultural drought, ignoring the differences in water sensitivity of crops at different growth stages. Bal et al. selected the temperature and precipitation data of 1636 meteorological stations in India, and calculated the yield loss of six rain-fed crops such as cotton and peanut based on the DSI index [46]. Hu et al. considered daily temperature and precipitation data from 1960 to 2015 to calculate the drought index for summer maize fertility and calculated the maize yield reduction based on the relationship between climatic yield and time trend yield [47]. In this study, the Jensen model (phase multiplication model) was used to solve the above problem and it had a good prediction of yield and could faithfully reflect the functional relationship between stage water consumption and final yield. By correlating the copula function with the Jensen model, the agricultural drought risk based on frequency analysis was proposed from the perspective of the correlation between drought frequency and

drought loss rate. The traditional qualitative assessment methods are mainly clustered to two categories: (1) drought risk assessment method based on probability and statistics theory [48,49], and (2) drought risk assessment method based on regional disaster system theory [50,51]. The disadvantages are that they are not based on the physical process of drought risk formation, cannot reflect the uncertainty and dynamics of complex drought risk systems, and are difficult to evaluate quantitatively. Therefore, this study proposes a drought risk assessment model based on the process of drought events and forms a set of technical processes for drought risk assessment. This method could intuitively reflect the law of agricultural drought loss under different return periods, and the drought return period evaluation method could quantitatively evaluate the regional drought process timely and effectively. Besides, the drought process evaluation index based on the percentage of precipitation anomaly was conducive to the realization of pre-disaster prediction and post-disaster rapid assessment, thus providing scientific knowledge for reasonable and effective disaster prevention and mitigation programs.

## 5. Conclusions

Taking rice as the reference crop in Zunyi City, agricultural drought risk assessment in multiple return periods was carried out. Based on the traditional qualitative assessment method, a drought risk assessment model based on drought event processes was proposed, and a set of drought risk assessment technical processes was formed. The main conclusions are as follows:

(1) By using the return period calculated by a copula function and the agricultural drought loss rate calculated by the Jensen model, the possible losses caused by different frequency droughts in a certain area under a certain drought resistance condition can be quickly estimated by constructing the relationship curve of drought risk mechanism. This method is helpful for competent authorities to make scientific decisions, respond in time, and formulate appropriate and effective countermeasures.

(2) Mapping the spatial distribution of drought losses under different return periods helps to compare the vulnerability between regions at the county level, so that the drought competent authorities can actively take defensive practices to mitigate drought risks. Therefore, this method broadens the technique and knowledge of risk quantification in drought risk assessment.

In the future, the overall drought assessment of regional agriculture will be further considered, including food and economic crops such as wheat, corn, and rape.

**Author Contributions:** Conceptualization, Funding acquisition, Resources, H.L.; Conceptualization, Writing—Original Draft preparation, Writing—Review and Editing, J.Y.; Supervision, Project administration, H.P.; Validation, Resources, J.L.; Data Curation, C.S.; Resources, S.J.L.; Software, Z.X.; Validation, C.J.; Methodology, L.S. All authors have read and agreed to the published version of the manuscript.

**Funding:** This research was funded by the National Natural Science Foundation of China (No. 52079052), the Key Science and Technology Project of Henan Province (No. 212102110032), the Major Science and Technology Innovation Project of Shandong Province (No. 2019JZZY010710), and the Fund of Innovative Education Program for Graduate Students at North China University of Water Resources and Electric Power, China (No. YK-2021-33).

**Institutional Review Board Statement:** Not applicable.

**Informed Consent Statement:** Not applicable.

**Data Availability Statement:** Data are contained within the article.

**Acknowledgments:** We fully appreciate the editors and all anonymous reviewers for their constructive comments on this manuscript. We would like to express our warm thanks for the English improvement to Chen Yingying from North China University of Water Resources and Electric Power.

**Conflicts of Interest:** The authors declare no conflict of interest.

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
