# Peer review of "A New Agricultural Drought Disaster Risk Assessment Framework: Coupled a Copula Function to Select Return Periods and the Jensen Model to Calculate Yield Loss"

_sustainability, doi:10.3390/su15043786_

Round 1

Reviewer 1 Report

1.      Modify the abstract for more result centric

2.      Briefly mention agricultural land in China affected by drought and drought frequency and crop loss mechanism in the introduction section.

3.      Check “thefocus” for correction line no. 45

4.      Rewrite the sentence “Up to now, drought disaster risk .. drought resilience” for better clarity line no. 50-51

5.      It is better to “drought” in place of “drought events” line no. 57

6.      The sentence “So the relationship 58 that returns period was equal to the reciprocal of drought frequency, and not applicable.” Line no. 58-59 Seems to be incomplete modify it for better suitability

7.      Figure 1 and 2 are ok

8.      The subheading “2.3.1. Drought classification based on Pa” will be to write as “Drought classification”

9.      Replace “drought process” with “drought stress” line no. 121

10.  It is better to replace “Extraction” with “analysis” in subheading 2.3.2.

11.  Replace “drought hazard” with “Drought impact” line no. 132

12.  Table 1, 2, 3 and 4 are ok, it is better to bold the first row in all table

13.  The method and material section is written nicely

14.  Table 5 and 6 are ok

15.  Figure 3 and 4 are ok but check figure 3 captions for correction

16.  Figure 5, 6 and 7 are ok

17.  The result section is written and presented nicely

18.  The discussion section is also written nicely and the interpretation of results in the discussion is ok

19.  Modify the conclusion section for better soundness

Reviewer 2 Report

The Manuscript ID Sustainability - 2202377 “A new agricultural drought disaster risk assessment framework: coupled Coupla function to select return periods and Jensen model to calculate yield loss s” describes a study filling a couple of useful purposes. The study provides useful scientific data and knowledge for reasonable and effective drought disaster prevention and mitigation in the area under study. The paper requires minor revisions.

Line 3: Is it Coupla or Copula, please clarify and revise throughout the manuscript.

Line 17: Replace “It proposed” with “This study proposes ……………

Line 18: Replace “framkwork” with “framework”

Lines 14 – 31: The abstract should ideally be structured according to the IMRaD format (Introduction, Methods, Results and Discussion). The methods are missing in the abstract.

Lines 74 – 89: This is not introduction but methodology of the some section of the study. Transfer whole of this section (line 74-89) to the methods and insert it in the appropriate section.

Lines 74 – 89: Here, just before the Objectives statement, there should be a concise Rationale statement (Why this specific study was needed?). What specific knowledge gap existed? Then please provide the objectives of the study.

Section 2.1: Replace “Study region overview” with “Description of the study area”

Lines 158 – 163: This is not methodology and should be transferred to the appropriate section or deleted.

Line 117: Insert “as described” between adopted and by.

Line 193: “include” should be “included”

Lines 193 – 194: What was the basis for the selection of the five major crops included for irrigation water allocation? Clarify.

What was the basis of using rice in this study and yet rice requires plenty of water as compared to other cereal crops. The authors should explain why they think the data obtained in this study using rice as a crop are not biased.

Line 229: Figure 3 has improper quality; probably its resolution should be improved.

Lines 378 – 391: These are not conclusions; conclusions are usually based on the objectives of the study. Please revise the conclusions.

Reviewer 3 Report

Dear Authors

I hope you are in good state of health. It is requested to revise the manuscript as per comments given. 

Regards
